# Patellofemoral Angle, Pelvis Diameter, Foot Posture Index, and Single Leg Hop in Post-Operative ACL Reconstruction

**DOI:** 10.3390/medicina59030426

**Published:** 2023-02-22

**Authors:** Ahmet Serhat Genç, Nizamettin Güzel

**Affiliations:** Department of Orthopedics and Traumatology, Samsun Training and Research Hospital, Samsun 55090, Türkiye

**Keywords:** Q angle, anthropometry, ACL reconstruction, foot posture, single leg hop test

## Abstract

*Background and Objectives:* Anterior cruciate ligament (ACL) injuries occur as a result of the deterioration of the static and dynamic stability of the knee. One of the structures involved in providing static stability is the patellofemoral angle (Q angle). The aim of this study was to investigate the relationships between Q angle, pelvis diameter, lower extremity length, and foot posture index (FPI) in patients who had undergone ACL reconstruction (ACLR) with the semitendinosus/gracilis (ST/G) technique on both the operated and non-operated sides. *Materials and Methods:* Twenty-five male recreational athletic patients between the ages of 18 and 35 who had undergone semitendinosus/gracilis (ST/G) anterior cruciate ligament reconstruction at least 6 months earlier were included in the study. Femur length, lower extremity length, pelvis diameter, and Q angle measurements, total foot posture index (FPI) scores, and single leg hop (SLH) and triple hop distance (THD) test results were determined on the operated and non-operated sides. *Results:* When the findings of the patients were evaluated statistically between the operated and non-operated sides, no significant differences were found in Q angle, femur length, and lower extremity length (*p* > 0.05). In terms of FPI scores, a significant difference was found only in the inversion/eversion of the calcaneus (CALC) parameter (*p* < 0.05). When the single hop test (SLHT) results were evaluated statistically on the operated and non-operated sides, the results were in favor of the non-operated side (*p* < 0.05). In the correlation analysis conducted for both the operated and non-operated sides, positive and significant correlations were found only between SLH and THD (*p* < 0.05). No significant difference was found in the other parameters. *Conclusions*: The fact that ST/G ACLR 6th month post-operative findings revealed similar results in Q angle, lower extremity length, and total FPI scores between the operated and non-operated sides showed that the 6-month process did not cause a difference in these parameters. However, it was found that the operated sides showed lower findings compared to non-operated sides for SLHTs, although these findings were within normal ranges in terms of the limb symmetry index.

## 1. Introduction

The anterior cruciate ligament (ACL) is one of the most important structures providing stabilization of the tibiofemoral joint, since it prevents anterior tibial translation and rotation in the knee [1]. ACL arthroscopic reconstruction is a procedure applied to restore anterior cruciate ligament function in individuals with anterior cruciate ligament deficiency and to reduce the risk of osteoarthritis and degeneration in other soft tissues of the knee joint that may occur in the future [2,3]. ACL injuries occur as a result of deterioration in tibiofemoral joint stability provided by the static and dynamic stability mechanisms of the knee [4]. One of the structures involved in providing static stability is the patellofemoral angle (Q angle) [5]. The Q angle is defined as the angle between a line drawn from the anterior superior iliac spine (ASIS) to the center of the patella and another line drawn from the center of the patella to the center of the tibial tubercle [6]. It is thought that when the Q angle exceeds the limit of 15–20°, it causes deterioration in the knee extensor mechanism and patellofemoral pain, with an increasing tendency of the patella to slide laterally [7]. Abnormally low values have also been associated with various problems [8]. In addition to causing injuries related to the knee, the Q angle can also be affected by many factors, such as femur length, pelvis width, and postural disorders [9]. When the gender factor is considered, it can be seen that pelvis width, which is one of the factors that changes the Q angle, is higher in women than in men [10,11].

Foot posture differences can lead to postural stability and musculoskeletal problems [12,13]. It is thought that the risk of injury is elevated, and sports performance is negatively affected, since structural deviations of the foot (pronation or supination displacement, high or low arch of the foot) may cause biomechanical deviations [14,15,16]. The association between foot morphology and lower extremity injuries is not clear; however, studies in the literature show both weak and strong correlations between arch structure and biomechanical characteristics of the lower extremity [17,18].

Single leg hop tests (SLHT) have become a widely adopted assessment tool in the rehabilitation of patients following anterior cruciate ligament (ACL) reconstruction, as well as a key factor in determining their readiness to resume sports participation. Researchers have stated that SLHTs are very important for measuring leg strength on a single joint, and they are commonly used to evaluate the functional states of individuals, identify asymmetries between the operated and non-operated sides, and follow developments in the limb [19,20,21].

Given all of these considerations, this study aimed to examine and evaluate the relationship between patellofemoral angle and foot posture index in patients who had undergone semitendinosus/gracilis (hamstring autograft) anterior cruciate ligament reconstructions (ACLR) and to compare surgically repaired knees with healthy knees.

## 2. Materials and Methods

### 2.1. Participants 

The subject group of the study consisted of 25 male recreational athletes between the ages of 18 and 35 who had undergone conventional ACLR with the semitendinosus/gracilis (ST/G) technique (Table 1). The inclusion criteria were as follows: patients who had a diagnosis of ACL rupture on one knee only; did not have comorbid meniscal, chondral, or other ligament injuries; did not have any other neuromuscular or musculoskeletal system injuries or a history of contralateral knee surgery or injury; and had undergone semitendinosus/gracilis (hamstring autograft) ACLR at least six months earlier. The optimal number of subjects to be included in the study was determined using the GPower 3.1.3. program. Before starting the study, the subjects were informed in detail and included in the study after signing an “Informed Consent Form”. 

### 2.2. Experimental Design

The study was evaluated as a retrospective cohort that included 25 males who had undergone the conventional ACLR (ST/G) technique by the same surgeon. The retrospective cohort part of the study included only post-operative 6th month Q angle, pelvis width, foot posture index (FPI), femur length, lower extremity length, single leg hop (SLH), and triple hop distance (THD) measurements. Additionally, prospective evaluations of patients undergoing ACL reconstruction included the assessment of Lysholm, Tegner, and International Knee Documentation Committee (IKDC) scores both prior to and at least 6 months following the operation. All measurements (Q angle, pelvis width, FPI, femur length, lower extremity length, single leg hop, and triple hop) were taken on the same day at the same hour (12:00–16:00). While taking the measurements, the single hop and triple hop test measurements were made after all the other tests were applied so that their findings would be correct against acute fatigue.

### 2.3. Procedures

#### 2.3.1. Hop Tests 

SLH: The patients stood on one foot on a line drawn to the floor, with the big toe touching the line. They were asked to jump as far as possible with one leg with their arms on both sides. The distance the patient jumped was determined by measuring the distance from the start line to the heel of the landing leg at the end of a single jump [22].

THD: In this test, the patients stood on one leg again. They jumped three times in a row, going as far as possible. The distance taken at the end of the three jumps was measured. The measurement was made again by measuring the distance from the start line to the heel of the landing leg at the end of three jumps [23].

#### 2.3.2. Anthropometric Measures 

Pelvis width was determined by measuring the distance between both the anterior and superior iliac spines in centimeters. Femur length was determined by measuring the distance between the trochanter major crest and the medial condyle in centimeters. The lengths of the leg and the distance between the anterior superior iliac spine (ASIS) or umbilical region and the medial malleolus were measured. All lengths were recorded in centimeters.

Q angle (quadriceps angle): This angle was measured from the right knee with the subject in the supine position on a horizontal table and the quadriceps muscle relaxed, with both lower extremities in full extension. The measurements were made using an Insize brand digital goniometer. Marks were placed on the anterior superior iliac spine, the center of the patella, and the tibial tubercle, and the midpoint of the goniometer was placed on the center of the patella. One arm of the goniometer was aligned to the ASIS point, while the other arm was aligned to the tibial tubercle point, and the Q angle was recorded in degrees [3].

FPI: It was first defined by Redmond et al. at a national podiatry conference in Australia. The original version of the index consisted of eight criteria, and it was referred to as the “FPI-8”. However, according to later studies, the last two criteria were found to be problematic, so the index was revised by deleting them. As a result, the “FPI-6” was developed. It is an assessment method with proven validity and reliability. It provides information about the general posture of the foot. The “FPI-6” consists of six criteria:(1)Talus head palpation (TH)(2)The curvature below and above the lateral malleolus (LATM)(3)Inversion and eversion of the os calcaneus (CALC)(4)Protrusion in the talonavicular joint area (TNJ)(5)Medial longitudinal arch alignment (MA)(6)Abduction and adduction of the forefoot when compared with the rearfoot (ABD)

Based on observation, each criterion is evaluated on five scales between −2 and +2 and scored accordingly. The total score (between −12 and +12) is the result of the entire evaluation. Positive values indicate pronation, and negative values indicate supination. A value between −5 and −12 shows severe supination of the foot, while a value between −1 and −4 shows supination. A value between 0 and +5 shows the normal position of the foot, a value between +6 and +9 shows pronation of the foot, and a value between +10 and +12 shows severe pronation of the foot [24,25].

### 2.4. Statistical Analysis

Statistical analyses were performed using the SPSS 21 software package. Descriptive statistics were used to present the results as mean and standard deviation. The normality assumption was assessed using the Shapiro–Wilk test, while Levene’s test was used to assess homogeneity. Paired sample *t*-tests were used to compare paired groups (i.e., the operated and non-operated limbs). Pearson’s correlation test was used to detect associations between parameters. Additionally, effect sizes were calculated using Cohen’s d effect size, which is calculated as (M2–M1)/SDpooled. Effect sizes were categorized as weak (d < 0.2), moderate (0.2 ≤ d < 0.5), or strong (d ≥ 0.5). Statistical significance was set at *p* < 0.05.

## 3. Results

When the lower extremity lengths and quadriceps angles of the operated and non-operated sides were evaluated, no significant differences were found between the Q angle (*p* = 0.668, 95% CI = −1.17–1.79), femur length (*p* = 0.515, 95% CI = −0.64–0.33), and lower extremity length (*p* = 0.904, 95% CI = −0.72–0.64) values (Table 2, Figure 1).

When the TFPI scores of the operated and non-operated sides were compared, no significant differences were found between the TH (*p* = 0.574, 95% CI = −0.19–0.11), LATM (*p* = 0.574, 95% CI = −0.11–0.19), TNJ (*p* = 0.425, 95% CI = −0.12–0.29), MA (*p* = 0.327, 95% CI = −0.04–0.12), ABD (*p* = 0.265, 95% CI = −0.34–0.10), and TFPI (*p* = 0.574, 95% CI = −0.56–0.32) values. On the other hand, statistical significance was found for CALC (*p* = 0.043, 95% CI = −0.31–−0.01) values (Table 3, Figure 2).

When the SLH (single leg hop) and triple hop distance test results of the operated and non-operated sides were evaluated, statistical significance was found for SLH (*p* = 0.004, 95% CI = −16.64–−3.60) and THD (*p* = 0.022, 95% CI = −41.99–−3.61) (Table 4, Figure 3).

When the correlations between pelvis diameter, Q angle, FPI, lower extremity length, and SLHT on the operated side were examined, positive correlations were found between TFAPI and ABD (r = 0.554), TFFPI and TNJ (r = 0.444), ABD and CALC (r = 0.520), TFAPI and TH (r = 0.646), TFAPI and LATM (r = 0.730), TFAPI and CALC (r = 0.713), and THD and SLH (r = 0.917). Negative and significant correlations were found between SLH and CALC (r = −0.475) and THD and CALC (r = −0.441) (Table 5).

When the correlations between pelvis diameter, Q angle, foot posture index, lower extremity length, and SLH tests on the non-operated side were examined, positive correlations were found between LATM and TH (r = 0.492), ABD and CALC (r = 0.485), CALC and TH (r = 0.517), CALC and LATM (r = 0.600), ABD and LATM (r = 0.510), TFAPI and TH (r = 0.729), TFAPI and LATM (r = 0.667), TFAPI and CALC (r = 0.794), TFAPI and ABD (r = 0.567), and THD and SLH (r = 0.867). Negative and significant correlations were found between TNJ and Pelvis D. (r = −0.403), SLH and CALC (r= −0.413), and THD and CALC (r = −0.427) (Table 6).

## 4. Discussion

The results of our study did not show any significant differences between the operated and non-operated sides in the 6th month post-operative Q angle, lower extremity length, or total FPI scores of patients who had undergone ST/G ACLR surgery. These results indicated that the operated side reached the healthy side in Q angle, lower extremity length, and FPI over a period of approximately 6 months. It was found that the operated sides showed lower results than the non-operated sides for SLHTs. However, when these results were evaluated in terms of the limb symmetry index (LSI), they were found to be within the normal range. Although significant differences were not found in the total FPI scores, significance was found only in the inversion/eversion of the calcaneus score between the operated and non-operated sides. The most important results in the correlation analyses evaluated on both the operated and non-operated sides was the negative and significant correlations between the CALC parameter in FPI and SLHTs.

Studies have been found in the literature that examined Q angles, SLHTs, and some physical characteristics in subjects after different ACL autograft methods. Dhillon et al. [26] did not find any significant differences in patients who had undergone ACLR with the patellar tendon graft method when compared with the control group in terms of pre-operative and post-operative Q angle (preop −13.86°, postop −12°). A Q angle of 8–15° is considered normal in men, while a Q angle of 12–19° is considered normal in women [27,28,29,30,31]. When Hertel et al. [32] compared the Q angles of 20 healthy subjects (11.84°) and 20 patients who had undergone ACLR (11.16°), they did not find any significant differences. An increase in this angle leads to uneven distribution of weight on the knee joint, exposing the medial or lateral knee joint compartments to more stress and joint disorders. Moreover, it may cause collapse in the medial arch of the foot because it creates an increase in joint pronation [7,33]. Increasing the subtalar will cause traction in the medial knee joint, lateral compression tension, supination in the transverse tarsal joint, and increased internal rotation-flexion-adduction angles in the hip joint. On the contrary, following an increased subtalar joint supination angle, there will be medial compression in the knee joint, lateral traction tension, pronation in the transverse tarsal joint, and increased external rotation-extension-abduction angles in the hip joint. In addition, an increase in the Q angle may cause a collapse in the medial arch of the foot [34]. In a study conducted in 2009, Barrios et al. showed that the subtalar joint supination angles were directly proportional to the Q angle and that the subtalar joint supination and tibial mechanical axis measurements were important in the estimation of varus stresses on the knee joint [35]. Although researchers have studied the norm values of the Q angle, there are also studies reporting that there may be differences in these angular values depending on certain factors, such as gender, age, and measurement position [29,36,37,38]. While some researchers have reported that the measurements on knee flexion according to knee extension position were not reliable [10,39], others have stated that there were no differences between Q angle measurements in extension and flexion of the knee and that the differences could be due to errors in determining measurement points or because the measurements were taken by different researchers [40]. When evaluated with all this information in the literature, the findings of our study that there were similar results on the operated and non-operated sides in terms of the 6th month Q angle indicate that the subjects had a good rehabilitation process after their ACLR, especially for the operated sides. It should also be considered that the 6-month process may lead to similar results in Q angles, depending on foot domination. Indeed, these results also show the main limitations of our study. The fact that no measurements were taken before ACLR has caused us not to show pre-operative and post-operative differences and to make interpretations based only on the literature. However, in the literature-supported findings, as in our study, the absence of any difference in Q angles after ACLR is thought to be due to the fact that ACLR does not include any procedure that would affect the patellofemoral angle. However, changes in the Q angle may occur when there are additional injuries that accompany the ACL tear. This situation may arise with detailed studies being carried out with a prospective design.

Besides knee injuries, the Q angle can be affected by many other factors, such as femur length, pelvis width, and posture disorders [9]. In their study, Murat et al. [41] found a negative weak correlation between Q angle and femur length, regardless of gender.

In addition, while it is claimed that high Q angle values are directly proportional to the width of the pelvis, some studies have not been able to confirm this result [27,28,42,43,44]. No significant correlation was found between pelvis width and Q angle in our study. This result also supports the literature information above, and it is thought that this may be due to the fact that our subjects were all male. As a matter of fact, it is known that women have higher pelvis widths and diameters than men due to their anatomical structures; therefore, studies on anthropometry have shown different levels of correlation in women compared with men. In a study conducted by Hertel et al. [32], the Q angle was found to be 12.7° in women and 10.2° in men, regardless of injury history, which showed that women had significantly higher Q angle results than men.

Another factor affecting the Q angle is ankle deformities [45]. Any injury or deformity in the foot or ankle disrupts the body’s biomechanics, starting from the knee, and if no precautions are taken, the problems in the body worsen [46]. Researchers have reported that with an increased Q angle, the foot tends toward pronation, and the amount of load carried on the medial side increases. Conversely, decreased Q angle causes supination of the foot and more load on the lateral side [33]. Considering that foot posture differences between the operated and non-operated sides may cause musculoskeletal problems, such as problems in the Q angle in FPI values after ST/G ACLR, no difference was found between the operated and non-operated sides in terms of total FPI scores. However, although both the operated and non-operated sides showed significant differences in the CALC parameter, both revealed findings close to pronation, but no significant correlation was found between the CALC results and the Q angles of the subjects. It is known that high-level athletes have flexible pronation in the calcaneum, and the deviation of the forefoot varies in almost all normal individuals. In addition, directions for palpation and curvature of the talus head are variable in almost all normal individuals. For this reason, it is important to evaluate kinematic analyses that will provide clearer findings instead of FPI in future studies.

The SLHT is commonly used after ACLR to assess functional performance [47,48,49,50,51]. SLHTs are frequently utilized in clinical practice to assess limb asymmetries between the operated and non-operated sides and to monitor the progress of lower extremity development following ACL reconstruction [19,20,21]. Studies on healthy individuals have reported that the difference between the limbs for SLH and THD tests between conventional SLHTs is, at most, 10–15% [52,53]. While one study found >90% similarity between the limbs in all of the participants that were tested with conventional SLHTs, another study conducted on healthy and athletic groups with a history of ACLR did not find limb asymmetry in conventional SLHTs [54,55]. In our study, it was found that although the operated sides showed much lower results than the non-operated sides, the results were still within the normal ranges in terms of LSI. Other important results of our study are the negative and significant correlations between the CALC parameter, which is an FPI score, and the SLH and THD tests. This shows that increased pronation in foot posture negatively affects performance in SLHTs. Indeed, since it is known that high Q angles correlate with feet that have high pronation [33], especially in high LSI rates that may occur in SLHTs after ACLR, the Q angle and FPI indices should be evaluated.

All our results show that the main limitation of our study is the lack of evaluation of healthy subjects as a control group. The fact that we did not have healthy subjects in our study made us unable to clearly explain the relationship between operated and non-operated sides in terms of all parameters.

## 5. Conclusions

The fact that the ST/G ACLR 6th month post-operative results were similar between the operated and non-operated sides in Q angle, lower extremity length, and total FPI scores showed that the 6-month-long process did not cause a difference in these parameters. However, it was found that the operated sides showed lower results than the non-operated sides for SLHTs, although these results were within normal ranges in terms of the limb symmetry index. Since it is thought that the non-operated sides may affect the results of the operated sides in patients who undergo ACLR, using healthy controls in future studies will enable us to obtain clearer results. In addition, applying SLH tests not only in the forward direction but multidirectionally and evaluating post-ACLR Q angles and FPI scores of subjects in future studies is important for seeing the outcomes more clearly.

## Figures and Tables

**Figure 1 medicina-59-00426-f001:**
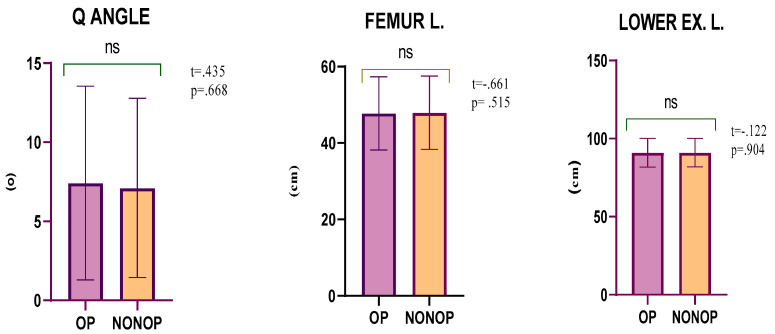
Comparison of lower extremity lengths and quadriceps angles of the operated and non-operated sides.

**Figure 2 medicina-59-00426-f002:**
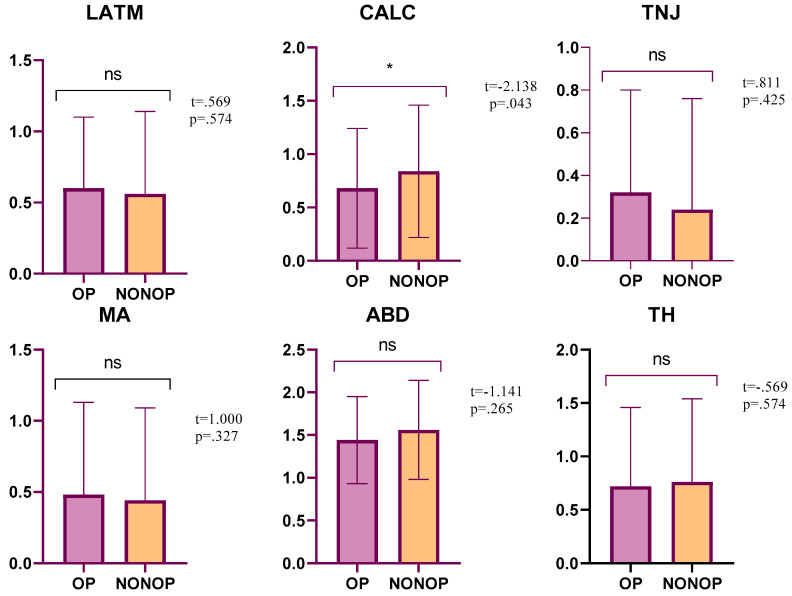
Comparison of TFPI scores of operated and non-operated sides. * *p* < 0.05.

**Figure 3 medicina-59-00426-f003:**
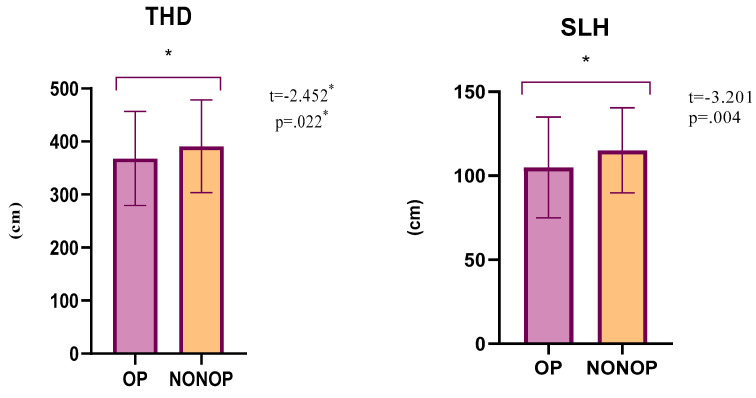
Comparison of TFPI scores of operated and non-operated sides. * *p* < 0.05.

**Table 1 medicina-59-00426-t001:** Descriptive data of the subjects (n = 25).

Patient Demographics	Mean	SD	Min	Max
Age (years)	29.28	6.86	18.0	35
Height (cm)	176.36	6.30	167.0	190.0
Weight (kg)	85.28	11.52	70.0	125.0
BMI (kg/m^2^)	27.36	2.64	22.60	34.63
Follow-up (months)	6.50	1.2	6	8
Pelvis D. (cm)	39.67	3.50	32.8	51.9
	R	L		
Operated Side (n/%)	15/60	10/40		
Dominant Side (n/%)	20/80	5/20		

SD standard deviation; Min minimum; Max maximum; BMI body mass index; R right; L left; CI confidence interval; Pelvis D. pelvis diameter.

**Table 2 medicina-59-00426-t002:** Comparison of lower extremity lengths and quadriceps angles of the operated and non-operated sides.

	OP	NONOP	*t*	*p*	ES	95% CI	
	Mean ± S.D	Mean ± S.D				LB	UB
Q angle (°)	7.42 ± 6.12	7.11 ± 5.67	0.435	0.668	0.05	−1.17	1.79
Femur L. (cm)	47.76 ± 9.56	47.91 ± 9.59	−0.661	0.515	0.02	−0.64	0.33
Lower Extremity L. (cm)	91.00 ± 9.13	91.04 ± 9.09	−0.122	0.904	0.00	−0.72	0.64

SD standard deviation; Min minimum; Max maximum; OP operated; NONOP non-operated; CI confidence interval; LB lower bound; UB upper bound; Q angle quadriceps angle; Femur L. femur length; Lower Extremity L. lower extremity length; ES Cohen’s d effect size.

**Table 3 medicina-59-00426-t003:** Comparison of TFPI scores of operated and non-operated sides.

	OP	NONOP	*t*	*p*	ES	95% CI	
	Mean ± SD	Mean ± SD				LB	UB
TH	0.72 ± 0.74	0.76 ± 0.78	−0.569	0.574	0.05	−0.19	0.11
LATM	0.60 ± 0.50	0.56 ± 0.58	0.569	0.574	0.07	−0.11	0.19
CALC	0.68 ± 0.56	0.84 ± 0.62	−2.138	0.043 *	0.27	−0.31	−0.01
TNJ	0.32 ± 0.48	0.24 ± 0.52	0.811	0.425	0.16	−0.12	0.29
MA	0.48 ± 0.65	0.44 ± 0.65	1.000	0.327	0.06	−0.04	0.12
ABD	1.44 ± 0.51	1.56 ± 0.58	−1.141	0.265	0.22	−0.34	0.10
TFPI	4.24 ± 1.98	4.36 ± 2.14	−0.569	0.574	0.06	−0.56	0.32

* *p* < 0.05; SD standard deviation; OP operated; NONOP non-operated; CI confidence interval; LB lower bound; UB upper bound; TH talar head palpation; LATM curves above and below the lateral malleolus; CALC inversion/eversion of the calcaneus; TNJ prominence in the region of the TNJ; MA congruence of the medial longitudinal arch; ABD abduction/adduction forefoot on rearfoot; TFPI total foot posture index; ES Cohen’s d effect size.

**Table 4 medicina-59-00426-t004:** Comparison of single leg hop and triple hop distance test results of operated and non-operated sides.

	OP	NONOP	LSI	*t*	*p*	ES	95% CI	
	Mean ± SD	Mean ± SD					LB	UB
SLH	104.96 ± 30.00	115.08 ± 25.32	90.73	−3.201	0.004 *	0.36	−16.64	−3.60
THD	368.20 ± 88.81	391.00 ± 87.60	94.48	−2.452	0.022 *	0.26	−41.99	−3.61

* *p* < 0.05; SD standard deviation; OP operated; NONOP non-operated; CI confidence interval; LB lower bound; UB upper bound; SLH single leg hop for distance; THD triple hop for distance; LSI limb symmetry index; ES Cohen’s d effect size.

**Table 5 medicina-59-00426-t005:** Correlation between pelvis diameter, Q angle, foot posture index, lower extremity length, and single hop test on the operated side.

	Pelvis D.	Q Angle	FL (cm)	LEL (cm)	TH	LATM	CALC	TNJ	MA	ABD	TFPI	SLH
Q Angle	−0.071											
FL (cm)	0.216	−0.098										
LEL (cm)	−0.162	−0.030	0.184									
TH	0.133	−0.257	−0.105	0.118								
LATM	0.058	0.314	−0.056	0.192	0.249							
CALC	0.207	0.147	−0.062	0.213	0.382	0.718 **						
TNJ	−0.261	0.134	0.228	−0.153	0.028	0.385	0.088					
MA	0.280	0.243	0.233	−0.014	0.118	−0.026	−0.133	0.289				
ABD	0.166	−0.071	0.187	0.144	0.233	0.395	0.520 **	−0.090	−0.035			
TFPI	0.193	0.119	0.109	0.147	0.646 **	0.730 **	0.713 **	0.444 *	0.389	0.554 *		
SLH	−0.221	−0.127	−0.094	0.265	−0.057	−0.298	−0.475 *	−0.008	0.082	−0.281	−0.276	
THD	−0.278	−0.134	−0.139	0.231	−0.026	−0.276	−0.441 *	−0.017	−0.048	−0.278	−0.294	0.917 **

* *p* < 0.05; ** *p* < 0.01; Q angle quadriceps angle; FL femur length; LEL lower extremity length; TH talar head palpation; LATM curves above and below the lateral malleolus; CALC inversion/eversion of the calcaneus; TNJ prominence in the region of the TNJ; MA congruence of the medial longitudinal arch; ABD abduction/adduction forefoot on rearfoot; TFPI total foot posture index; Pelvis D. pelvis diameter; SLH single leg hop for distance; THD triple hop for distance.

**Table 6 medicina-59-00426-t006:** Correlation between pelvis diameter, Q angle, foot posture index, lower extremity length, and single hop test on the non-operated side.

	Pelvis D.	Q Angle	FL (cm)	LEL (cm)	TH	LATM	CALC	TNJ	MA	ABD	TFAPI	SLH
Q Angle	−0.278											
FL (cm)	0.225	−0.031										
LEL (cm)	−0.193	0.167	0.146									
TH	0.145	−0.088	−0.009	0.196								
LATM	0.151	0.237	−0.030	0.224	0.492 *							
CALC	0.282	0.029	0.039	0.317	0.517 **	0.600 **						
TNJ	−0.403*	0.305	0.046	0.024	0.045	−0.049	−0.005					
MA	0.279	0.131	0.207	0.025	0.217	−0.018	0.180	0.167				
ABD	0.015	0.094	0.225	0.129	0.308	0.510 **	0.485 *	−0.049	−0.127			
TFAPI	0.123	0.227	0.123	0.252	0.729 **	0.667 **	0.794 **	0.329	0.420 *	0.567 **		
SLH	−0.183	0.135	−0.203	0.174	0.081	−0.017	−0.413 *	0.250	−0.129	−0.302	−0.171	
THD	−0.311	0.151	−0.180	0.180	0.084	0.030	−0.427 *	0.340	−0.226	−0.314	−0.158	0.867 **

* *p* < 0.05; ** *p* < 0.01; Q angle quadriceps angle; FL femur length; LEL lower extremity length; TH talar head palpation; LATM curves above and below the lateral malleolus; CALC inversion/eversion of the calcaneus; TNJ prominence in the region of the TNJ; MA congruence of the medial longitudinal arch; ABD abduction/adduction forefoot on rearfoot; TFPI total foot posture index; Pelvis D. Pelvis diameter; SLH single leg hop for distance; THD triple hop for distance.

## Data Availability

The datasets used and/or analyzed during the current study are available from the corresponding author on reasonable request.

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
