# Peer review of "Patellofemoral Angle, Pelvis Diameter, Foot Posture Index, and Single Leg Hop in Post-Operative ACL Reconstruction"

_medicina, 2023, doi:10.3390/medicina59030426_

Round 1
Reviewer 1 Report
The paper's title is far too long because it is not the result of a priority determination nor a synthetic thought. A title must be a clear message of what was discovered or measured, but not too detailed in terms of variables. For instance, the title could be Patella Femur Angle, Pelvis Diameter, Foot Posture Index and Single Leg Hop in Pre- and Post-operative ACL Reconstruction.
Table 1 is all wrong (Height and weight!!). Please be more careful before sending a paper to review.
Table 5 and table 6n should be merged into ONE table with both correlation coefficients of operated and non-operated limbs in the same cell for easy comparison.
On page 8, where one reads now "...study did not show significance between..." should be "...study did not show any significant difference between..."
Please check that acronyms are ALWAYS presented with the whole words the first time they are used. I fear a few are not introduced properly. Check the same applies to figures and tables, as self-contained units.
Author Response
Thank you very much for your contribution to our research. Undoubtedly, your contributions were very important to the improvement of our research. I'll try to respond to all the revisions you've given below.
The paper's title is far too long because it is not the result of a priority determination nor a synthetic thought. A title must be a clear message of what was discovered or measured, but not too detailed in terms of variables. For instance, the title could be Patella Femur Angle, Pelvis Diameter, Foot Posture Index and Single Leg Hop in Pre- and Post-operative ACL Reconstruction.
Response: The title has been revised as you said.
Table 1 is all wrong (Height and weight!!). Please be more careful before sending a paper to review.
Response: Table 1. has been revised as you said.
Table 5 and table 6n should be merged into ONE table with both correlation coefficients of operated and non-operated limbs in the same cell for easy comparison.
Response: We tried merging as you said but it became harder to read as there are too many parameters. In addition, due to the excess of parameters, the numbers have become very small.
On page 8, where one reads now "...study did not show significance between..." should be "...study did not show any significant difference between..."
Response: This sentence has been revised as you said.
Please check that acronyms are ALWAYS presented with the whole words the first time they are used. I fear a few are not introduced properly. Check the same applies to figures and tables, as self-contained units.
Response: We did all the checks as you said. Thank you for correcting our mistake.

Reviewer 2 Report
Interest effort trying to evaluate measurements of Q angle, and foot posture, as static measurements. These measurements are DIFFERENT, related to single or triple hop tests that are dynamic ones.
Measurements of FPI-6 is a weak area of the manuscript, since high level athletes have flexible pronation of the calcaneum and deviation of the forefoot has variation in almost all normal individuals. This is NOT mentioned in discussion. The variability of static foot measurements makes the test weak. This test CANNOT be an indication for the performance of the athlete after ACL reconstruction. Referrals on palpation of the talus head and curvatures are variable in almost all normal individuals.
Authors must comment in discussion the reason that Q angle could be different between operated and non operative sides, after the procedure! Of course no differences will be found.
In discussion their initial statement . These results indicated that the non-operated side reached the healthy side in Q angle, lower extremity lengths and FPIs in a period of approximately 6 months must be explained properly.
A WEAK point is the absence of any mention to measurements IKDC that they report in the design.
Author Response
Response to reviewer 2
Thank you very much for your contribution to our research. Undoubtedly, your contributions were very important to the improvement of our research. I'll try to respond to all the revisions you've given below.
Interest effort trying to evaluate measurements of Q angle, and foot posture, as static measurements. These measurements are DIFFERENT, related to single or triple hop tests that are dynamic ones.
Response: We are grateful to you for finding our research interesting. Also, thank you very much for your contribution to our research. We've tried to answer your vocabulary corrections below. Kind regards
Measurements of FPI-6 is a weak area of the manuscript, since high level athletes have flexible pronation of the calcaneum and deviation of the forefoot has variation in almost all normal individuals. This is NOT mentioned in discussion. The variability of static foot measurements makes the test weak. This test CANNOT be an indication for the performance of the athlete after ACL reconstruction. Referrals on palpation of the talus head and curvatures are variable in almost all normal individuals.
Response: We have added to the discussion the explanation that FPI-6 is a weak area of research as you said.
Authors must comment in discussion the reason that Q angle could be different between operated and non operative sides, after the procedure! Of course no differences will be found.
Response:We have revised and added as you said.
In discussion their initial statement . These results indicated that the non-operated side reached the healthy side in Q angle, lower extremity lengths and FPIs in a period of approximately 6 months must be explained properly.
Response: We added general comment in initial part of discussion.
A WEAK point is the absence of any mention to measurements IKDC that they report in the design.
As you said, we didn't talk much about tegner, lyhsolm and IKDC scores in the discussion. However, since our current research hypothesis focuses on other parameters and our IKDC scores were already known in the literature, we did not consider adding them. In addition, when we added, we were afraid that the discussion would get too long and tire the reader. We will definitely take what you say into consideration in our future research. With my most sincere regards.
